# A chemiresistive-potentiometric multivariate sensor for discriminative gas detection

Hong Zhang[1], Zuobin Zhang[1], Zhou Li [1], Hongjie Han[1], Weiguo Song[1] & Jianxin Yi [1]✉

Highly efficient gas sensors able to detect and identify hazardous gases are crucial for numerous applications. Array of conventional single-output sensors is currently limited by problems including drift, large size, and high cost. Here, we report a sensor with multiple chemiresistive and potentiometric outputs for discriminative gas detection. Such sensor is applicable to a wide range of semiconducting electrodes and solid electrolytes, which allows to tailor and optimize the sensing pattern by tuning the material combination and conditions. The sensor performance is boosted by equipping a mixed-conducting perovskite electrode with reverse potentiometric polarity. A conceptual sensor with dual sensitive electrodes achieves superior three-dimensional (sub)ppm sensing and discrimination of humidity and seven hazardous gases (2-Ethylhexanol, ethanol, acetone, toluene, ammonia, carbon monoxide, and nitrogen dioxide), and enables accurate and early warning of fire hazards. Our findings offer possibilities to design simple, compact, inexpensive, and highly efficient multivariate gas sensors.

Sensitive and discriminative detection of low-concentration gases is of great significance for a wide range of applications in public safety, environmental protection, health diagnostics, industry, etc[1-4]. Conventional gas analytical techniques such as gas chromatography and mass spectrometry allow for reliable detection and identification of trace-level gases. However, the inherent bulkiness, inconvenience, and high cost severely hamper their extensive applications[5,6]. Gas sensors are promising alternatives due to advantages such as low cost, small size, and ability for real-time monitoring[7-9]. Unfortunately, state-of-the-art gas sensors suffer a great deal from a high cross sensitivity and poor selectivity, and stand-alone sensors fail in gas recognition[10]. Sensor arrays (i.e., electronic noses), mostly based on a simple combination of classic single-output sensors, could possibly discriminate gases[11-13]. Nevertheless, the practical application of arrays has been seriously limited by the long-standing drift problem[14]. Moreover, using a larger number of standalone sensors in arrays for higher-order sensing and better accuracy is associated with a rise of device size

and cost, which is unfavorable for practical applications especially when portability and miniaturizability are demanded[15-17]. Achieving high sensing efficiency with minimized sensor number is imperative but challenging.

Multivariable sensor, which outputs multiple response signals, has attracted tremendous attention as a cost-effective and compact alternative to sensor arrays[18-20]. The different responses, being partially or fully independent, arise typically from the same sensing material, which favors not only drift alleviation but also size and cost reduction. The number of independent outputs in a multivariable sensor, i.e., the dimension of dispersion, plays a critical role in determining the discriminative power[18]. Existing multivariable sensors mostly possess two-dimensional (2D) dispersion (Supplementary Table 1). To increase the applicability, it is crucial that the sensor dispersion has a highest possible dimension. More importantly, the transducers combined should desirably be simple, low cost, and miniaturizable, and have good compatibility. Chemiresistive (C) and potentiometric (P) are two most widely

[1]State Key Laboratory of Fire Science, Department of Safety Science and Engineering, University of Science and Technology of China, Hefei, Anhui 230026, PR China. ✉e-mail: yjx@ustc.edu.cn

studied gas sensor transducers, both meeting all these requirements[21,22]. So far, the C signal has been combined for multivariate sensing with some others like volume, capacitance, work function, optical, and mass (Supplementary Table 1)[23–30]. Nevertheless, the counterpart transducers are often associated with high cost, limited applicability, and/or complexity. The P transducer is highly compatible with the C one in many aspects including materials and working conditions, but has been barely studied for multivariate gas sensing. A chemiresistive-potentiometric (C-P) multivariable sensor is envisioned to be a simple and cheap solution to discriminative gas detection for widespread applications.

In this work, we report a multivariate sensor with simple C and P outputs. The multivariate sensing is operable for sensing electrodes (SEs) of various n- or p-type semiconductor materials and on different solid electrolytes over a wide temperature range. By combining multiple SEs and replacing conventional Pt counter electrode (CE), the platform is able to achieve three-dimensional (3D) or higher response dispersion, enabling to tune or enhance the sensing performance while reducing the sensor cost. Further, a mixed conducting perovskite oxide, $Ba_{0.5}Sr_{0.5}Co_{0.8}Fe_{0.2}O_{3-\delta}$ (BSCF), which exhibits p-type C sensing and unusual P response polarity, is explored as a high-performance SE. Pairing BSCF with a regular SE like $SnO_2$ significantly boosts the P performance owing to their opposite response sign. The thus-obtained dual-SE sensor can detect

and discriminate between humidity and seven exemplary hazardous gases at a (sub)ppm-level, and achieves accurate and early warning of fire. The multivariate sensor offers numerous opportunities to tailor the sensing patterns suited to various applications by optimizing the materials and condition combinations. Our findings pave the way for the design of simple, low-cost, and highly efficient C-P multivariate sensors.

## Results

### Multivariable sensor with single SE

We first exemplify the C-P gas sensing with a single SE deposited on a $Ce_{0.8}Gd_{0.2}O_{1.9-\delta}$ (GDC) solid electrolyte substrate, as schematically depicted in Fig. 1a. Porous hollow nanofibers of $SnO_2$, a model n-type semiconductor gas sensing material, are used as the SE (see Supplementary Figs. 1 and 2 for characterization of the sensor and materials). The C signal results from the electrical resistance change due to reaction of analyte gas with charged oxygen adsorbates on the SE surface, while the P one ($SnO_2$-Pt) corresponds to variation of the electrode potential that arises mainly from electrochemical reactions at the gas/SE/electrolyte three-phase boundary (TPB). Figure 1b-c and Supplementary Fig. 3 display typical dynamic response curves of $SnO_2$. Seven representative hazardous or fire signature gases are chosen, including four volatile organic compounds (VOCs), 2-Ethylhexanol (2-EH), ethanol, acetone, and toluene, and three inorganic gases,

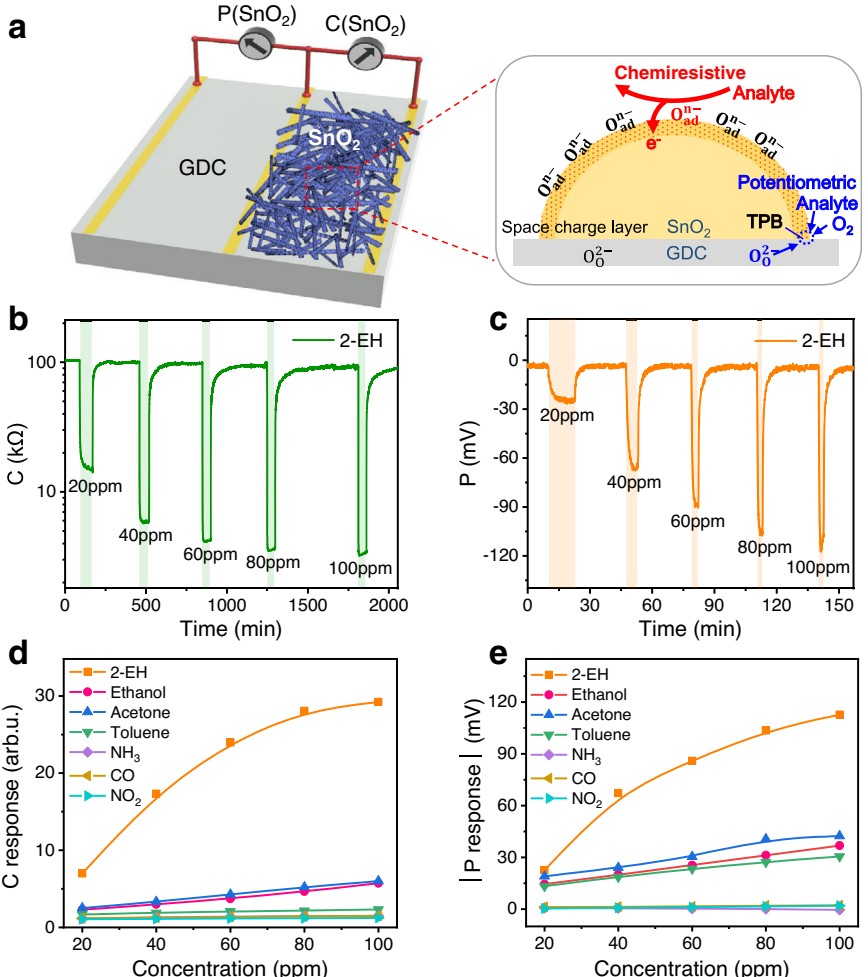

**Fig. 1 | Schematics and sensing properties of the multivariable sensor with SnO₂ nanofiber SE. a** Overview of the proposed multivariable sensor. The marked rectangle area shows the chemiresistive-potentiometric sensing mechanism. **b** Chemiresistive and **c** potentiometric (vs Pt CE) response/recovery transients of

the multivariable sensor in different concentrations of 2-EH. **d** Chemiresistive and **e** potentiometric response toward 2-EH, ethanol, acetone, toluene, NH₃, CO, and NO₂ as a function of gas concentration. Sensor operating temperature is 400 °C. Source data are provided as a Source Data file.

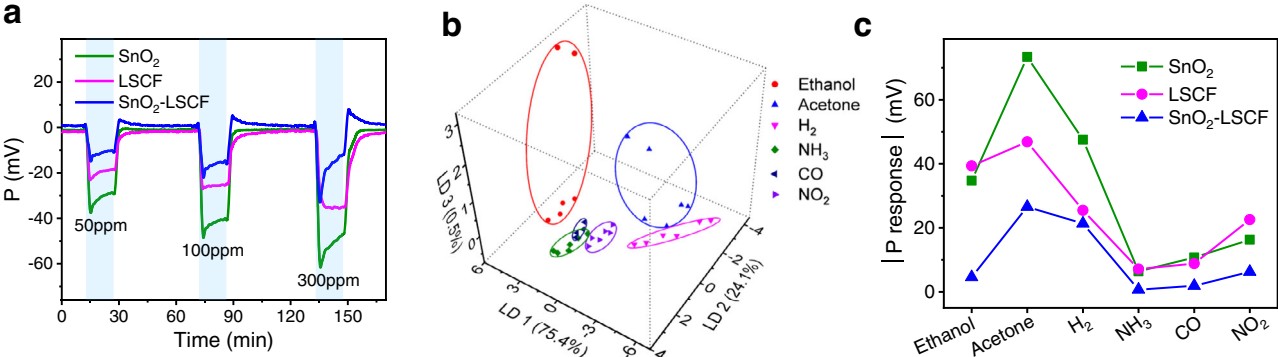

**Fig. 2 | Sensing properties of the dual-SE multivariable sensor with SnO₂ nanofibers and LSCF nanoparticles. a** Potentiometric response/recovery transients of SnO₂ and LSCF in different concentrations of H₂, where their signal difference gives the SnO₂-LSCF signal. The green, pink, and blue solid lines represent the P (SnO₂ vs Pt CE), P (LSCF vs Pt CE), and P (SnO₂ vs LSCF) signals, respectively. **b** LDA pattern recognition of 6 gases in the concentration range from 50 to 300 ppm using the SnO₂-LSCF C-P sensor. LD1, LD2, and LD3 represent first, second, and third discriminant function, in whose projection direction the data variances account for 75.4%, 24.1%, and 0.5% of the total variance, respectively. **c** Potentiometric response values of SnO₂, LSCF, and SnO₂-LSCF to 100 ppm various gases. Sensor operating temperature is 400 °C. Source data are provided as a Source Data file.

ammonia (NH₃), carbon monoxide (CO), and nitrogen dioxide (NO₂). Upon exposure of the device to the reducing (oxidizing) analyte gas, both the resistance and the potential (vs Pt CE) decrease (increase) substantially, which is typical n-type chemiresistive and non-Nernstian gas sensing behavior, respectively[31,32]. The responses tend to increase with increasing gas concentration and with decreasing temperature (Fig. 1d–e and Supplementary Fig. 4). SnO₂ exhibits high C and P responses to 2-EH, a representative indoor air pollutant and signature of explosives and early electrical fires[7,33–35], while responses to inorganic gases NH₃, CO, and NO₂ are rather low. For other VOCs, the P responses are moderate while the C responses are much smaller than that for 2-EH. The signals differ distinctly in the dynamic response and recovery behavior, which agrees well with the different sensing mechanisms, thereby suggesting that they are independent of each other (Supplementary Figs. 5 and 6). Considering that humidity is often an interference for gas sensors, the effect of humidity is also examined. The baseline resistance and potential as well as the responses decrease significantly with increasing relative humidity (RH) (Supplementary Fig. 7). In order to assess the discriminative capability of the single-SE C-P sensor, linear discriminant analysis (LDA) was performed on the response data of the gases (Supplementary Fig. 8). The gases are distinctly separated into four groups, i.e., 2-EH, other VOCs, inorganic gases, and humidity with an overall recognition accuracy of 52.1%. These results demonstrate the independence of the two outputs and the discriminative power of the C-P multivariable sensor, contrast to its single-output (either C or P) counterpart.

The C-P gas sensing is also realized on other solid electrolytes, including 8 mol% yttria-stabilized zirconia (YSZ), Er₀.₄Bi₁.₆O₃-δ (ESB), and Na₃Zr₂Si₂PO₁₂ (NASICON) (see Supplementary Figs. 9 and 10 for characterization of the electrolytes). The ion conductivity of the solid electrolytes largely determines the measurable temperature range of the P response, and higher ion conductivity favors lower temperature operation. The lowest operable temperature is around 400 °C for YSZ and GDC with relatively low ion conductivity, and extends down to 300 °C and 200 °C for the highly ion conductive ESB[36] and NASICON[37], respectively (Supplementary Fig. 11). Thus, the operable temperature range for the P output covers 200-500 °C, matching well with that of the C one. Moreover, the C-P sensor can be readily applied to various semiconductor oxides of different type, composition, and morphology, such as n-type ZnO nanofibers, p-type Fe-doped NiO nanofibers, and p-type La₀.₈Sr₀.₂Cr₀.₅Fe₀.₅O₃-δ (LSCF) nanoparticles (Supplementary Figs. 12–16).

## Multivariable sensor with multiple SEs

Like most conventional P gas sensors, the above single-SE sensors use highly conductive Pt as CE, which does not generate C response. Moreover, Pt is scarce and expensive, raising the sensor cost. Hence, a sensitive semiconductor material in place of Pt as CE may not only provide an extra C output but also serve as an additional SE, which will increase the sensing capability and reduce the cost. To validate this hypothesis, we prepare a planar C-P sensor by pairing SnO₂ with LSCF (SnO₂-LSCF), which is free of Pt CE (Supplementary Fig. 17). This sensor, consisting of the same number of electrodes as the Pt CE-based single-SE C-P sensor with two readouts, indeed generates 3 different outputs, i.e., C(SnO₂), C(LSCF), and P(SnO₂-LSCF). Although the P response is reduced by the pairing due to the same response polarity of the SEs (Fig. 2a), the dual-SE C-P sensor accomplishes 3D response (Fig. 2b), resulting in better discrimination accuracy (91.2%) than the single-SE ones (76.5% for SnO₂ and 88.2% for LSCF). Moreover, the discrimination performance is equivalent or even superior to that of arrays consisting of three single-output C/P sensors of SnO₂ and LSCF (Supplementary Table 2). Therefore, increasing the number of outputs by replacing Pt CE with another SE does improve the discriminative power. Further, more powerful discrimination is feasible by using more SEs on the same substrate. In principle, N different SEs yield (2N-1) independent outputs, i.e., N chemiresistive and (N-1) potentiometric ones. As an example, we obtain 7 outputs with 4 SEs based on Fe-doped NiO nanofibers (Supplementary Fig. 18), which are the basis for 7D response.

For a planar-type P sensor, which favors miniaturization due to the simple structure with exemption from the necessity of reference air, all dissimilar electrodes are exposed and thus can respond to the analyte gas. Under typical sensing conditions in ambient atmosphere, an analyte gas other than oxygen usually results in non-Nernstian P response, which has a negative (positive) sign for reducing (oxidizing) gases. This behavior can be attributed to generation of mixed potential and change of electron concentration (for semiconductors)[38], and holds generally for various types of materials including n- or p-type semiconductors and composites. Therefore, responses from different SEs have the same polarity and partially counteract each other, reducing the P response of the sensor (Fig. 2a and Supplementary Fig. 19). As each SE has its own sensing pattern, the change in the sensor response due to the SE pairing would differ in the magnitude for different gases, thereby varying the overall sensing pattern of the sensor (Fig. 2c). Despite the reduced P response, this change can be taken as an advantage, enabling to tailor or optimize the sensing pattern for

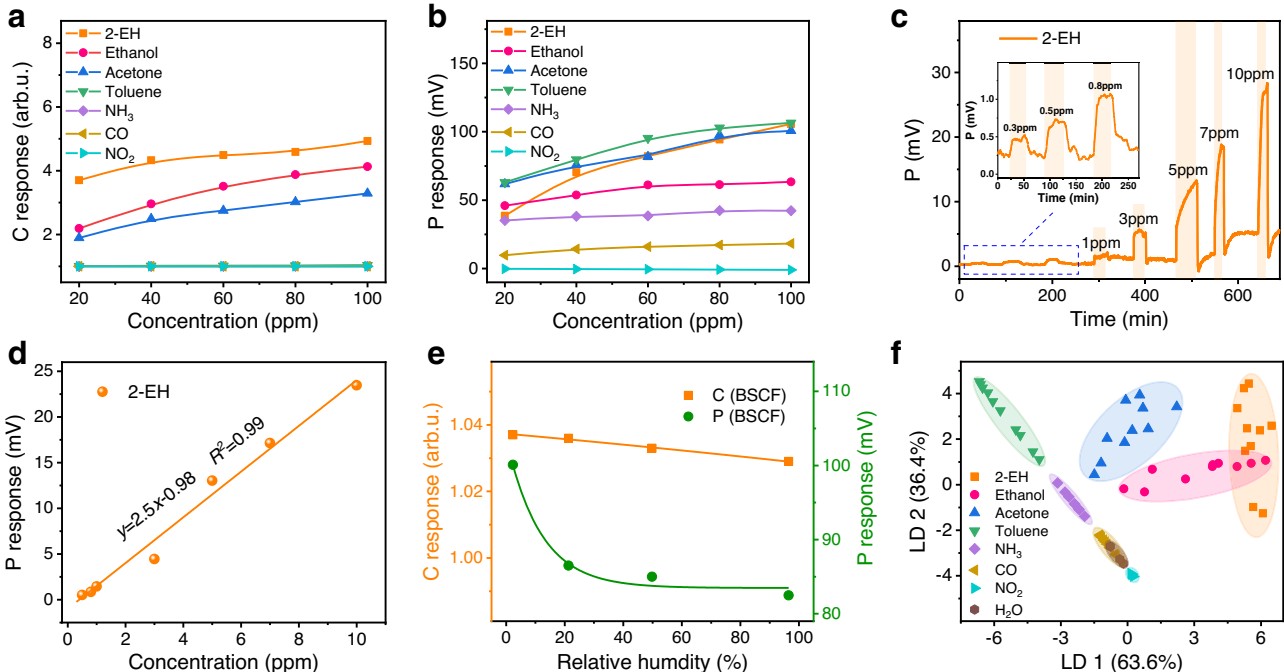

**Fig. 3 | Multivariable sensing performance of BSCF at 400 °C. a** Chemiresistive and **b** potentiometric (vs Pt CE) response of BSCF toward 7 target gases as a function of the gas concentration. **c** Dynamic potentiometric response curve for different 2-EH concentrations in the range from 0.3 to 10 ppm. Inset in (c) is a zoom of the sub-ppm (0.3-0.8 ppm) responses. **d** Linear relationship between the P response and 2-EH concentration. The solid line represents a linear fit. **e** Chemiresistive and potentiometric responses of BSCF to 100 ppm toluene under different relative humidity. **f** LDA pattern recognition of 7 gases in the concentration range from 20 to 100 ppm and relative humidity from 2.3% to 96.5% using the BSCF C-P sensor. The data variances in the projection direction of LD1 and LD2 account for 63.6% and 36.4% of the total variance, respectively. Source data are provided as a Source Data file.

better gas recognition or discrimination, as evidenced by the improved discrimination of the SnO2-LSCF C-P sensor. This feature contrasts the case for arrays where the overall pattern is unalterable.

**Mixed-conducting perovskite oxides as high-performance SE**

To maximize the discriminative power and efficiency, it is essential that every SE in a C-P sensor possesses high sensing performance and in the meantime the sensing patterns of the SEs differ as much as possible from each other. To this end, the SEs should differ greatly in the materials (such as compositions and morphology), properties, and even the sensing mechanism[39]. Most SEs studied so far are electronic conductors, which have almost the same C or P sensing mechanisms, undesired for the diversity of the response characteristics and sensing pattern. Seeking a high-performance SE of different mechanism is vital but also challenging. Mixed ionic-electronic conductors (MIECs) have excellent electrocatalytic activities due to the ability to conduct both oxide ion and electron (hole), and thus have been widely studied for fuel cells, gas separation membranes, and electrocatalysis[40]. Recently, we found that some MIECs exhibit an ideal unity of excellent potential response and an opposite polarity to usual one, which allows to substitute the expensive Pt electrode while greatly enhancing the performance of conventional P sensor[38]. This unusual behavior is attributed to a surface electrostatic mechanism rather than the traditional ones. In addition, MIECs are mostly p-type semiconductors with potentials for chemiresistive gas sensing. Therefore, MIECs could be a high-performance SE candidate for the C-P sensor.

As a demonstration, we measure the C-P response of BSCF, a classic MIEC perovskite oxide with superior electrocatalytic activities[41]. To our best knowledge, both the C and P gas sensing properties of BSCF remain unexplored yet. Figure 3 displays gas sensing properties of BSCF and Supplementary Figs. 20–21 show the characterization results. BSCF exhibits typical p-type chemiresistive behavior,

responding sensitively to 2-EH, ethanol, and acetone but almost negligibly to the others (Fig. 3a and Supplementary Fig. 22). The C responses of BSCF are lower than those of SnO2. By contrast, the P response behavior is distinctly different: pronounced responses are observed for most gases particularly the VOCs (Fig. 3b). For example, the response amounts to 105.7 mV for 100 ppm 2-EH. In addition, the P responses to the gases differ greatly, which is beneficial to the gas discrimination. More importantly, their signs are opposite to those of conventional materials, that is, the potential increases (decreases) for reducing (oxidizing) analyte gases (Supplementary Fig. 22). The P response rises with increase of gas concentration and decrease of temperature (Fig. 3b and Supplementary Fig. 23). Encouraged by the excellent P sensing performance of BSCF to VOCs, we explore the response for low concentrations. As shown in Fig. 3c, BSCF exhibits distinct response to 2-EH down to a concentration of 0.3 ppm. A large sensitivity of 2.5 mV/ppm is obtained, which results in a calculated limit of detection (LOD) as low as 26.6 ppb (Fig. 3d). Furthermore, the anti-humidity property of BSCF is also tested. As the RH in the base gas increases, both the base resistance and voltage increase slightly (Supplementary Fig. 24). The C response is almost unchanged in the whole RH range of 2.3%–96.5%. The P response decreases moderately by 13.5% in the lower RH range of 2.3%–21.3%, but only slightly by 4.7% within the higher RH range of 21.3%–96.5% that is frequently encountered in daily environments (Fig. 3e). Contrast to many regular semiconductors that are pronouncedly susceptible to humidity, MIEC BSCF possesses much better humidity resistance.

The very different C and P response behavior of BSCF is an advantage for gas discrimination. LDA plot for the BSCF-based single-SE C-P sensor shows well-separated clusters for the 8 gases, except slight overlapping of 2-EH and ethanol (Fig. 3f). A recognition accuracy of 84.9% is achieved, much better than that of the classic SnO2. Therefore, BSCF can be a powerful electrode for C-P sensor.

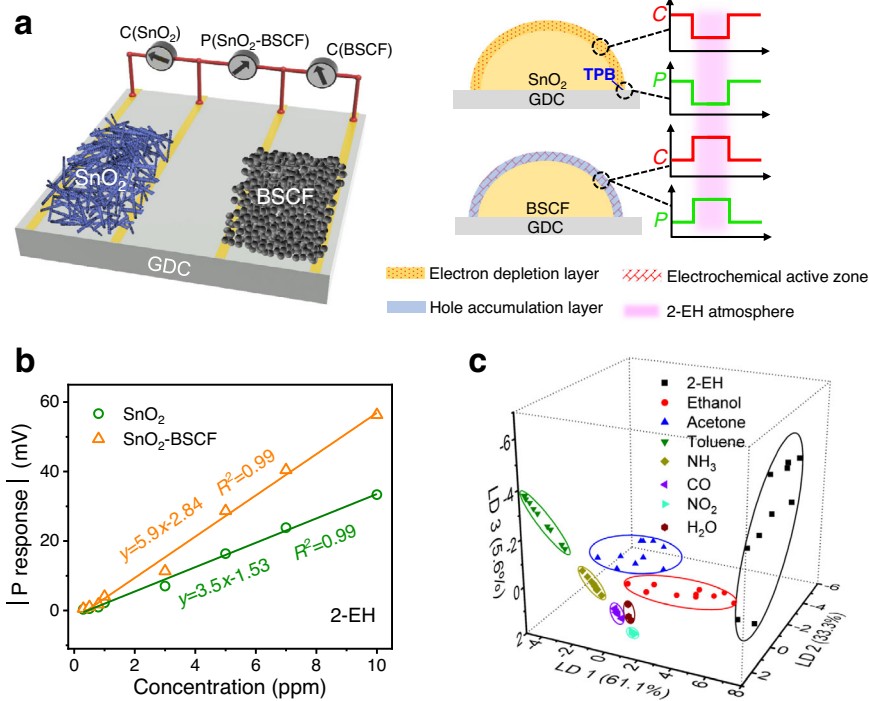

**Fig. 4 | Schematics and sensing properties of the dual-SE multivariable sensor with SnO₂ nanofibers and BSCF nanoparticles. a** Schematic illustration of the sensor configuration and sensing mechanism. In the presence of 2-EH analyte, the C and P signals of the SnO₂ electrode descend, resulting from reactions on the material surface and at the SnO₂/GDC/2-EH three-phase boundary (TPB), respectively, while those of the BSCF electrode both ascend due to reactions on the material surface. **b** Linear relationship between potentiometric responses and 2-EH concentration (0.3-10 ppm). Solid lines represent linear fits. **c** Pattern recognition of 7 gases in the concentration range from 20 to 100 ppm and relative humidity from 2.3% to 96.5% based on LDA using the SnO₂-BSCF C-P sensor. The data variances in the projection direction of LD1, LD2, and LD3 account for 61.1%, 33.3%, and 5.6% of the total variance, respectively. Sensor operating temperature is 400 °C. Source data are provided as a Source Data file.

## Enhancing the multivariable sensing performance

Large response, high sensitivity, and low LOD are particularly important for reliable earlier detection of hazardous or fire signature gases. The unusual reverse polarity of the P response for BSCF is a unique advantage, which can be utilized to enhance the performance of sensors with conventional SEs[38]. To exemplify, we pair BSCF with SnO₂ to construct a sensor with dual SEs of opposite P polarity (SnO₂-BSCF) (Fig. 4a and Supplementary Fig. 25). In such way, the P response and sensitivity of the SnO₂-BSCF sensor equal the sum of those for the corresponding single-SE sensors (all in absolute values), respectively (Supplementary Fig. 26). Compared with SnO₂-Pt sensor, SnO₂-BSCF manifests much larger P response values. For instance, the response amounts to 212.1 mV for 100 ppm 2-EH, corresponding to an increase of 47.4% (Supplementary Fig. 27). The sensitivity grows by 68.5% to 5.9 mV/ppm for 2-EH (Fig. 4b), and by 411% to 4.6 mV/ppm for toluene (Supplementary Fig. 28). Owing to the enhanced sensitivity, the calculated LOD achieves 8.1 ppb and 55.9 ppb for 2-EH and toluene, respectively. The LOD of P(SnO₂-BSCF) is all at the sub or low ppm level (Supplementary Fig. 29 and Supplementary Table 3), significantly lower than the occupational exposure limits[42] (except NO₂). Dynamic transients of the sensor reveal that the P signals are highly repeatable and fully recoverable (Supplementary Fig. 30). To examine the reliability of our sensor test and stability of materials, we intermittently measure the P responses to 2-EH over a period of 52 days, during which data for plotting the 3D response pattern are collected. Results reveal that the response fluctuates slightly without appreciable degradation, indicating excellent response reproducibility and sensor stability over the test cycle (Supplementary Fig. 31). The performance enhancement by pairing also favors the capability of gas discrimination and recognition. Figure 4c shows the LDA plot of SnO₂-BSCF for 7 different gases and humidity. Clearly, well-separated clusters of the analytes are

obtained. The SnO₂-BSCF C-P sensor achieves a high recognition accuracy of 97.3%, outperforming conventional 3-sensor arrays of SnO₂ and BSCF (CCP, CPP) and even the 4-sensor array with one more output (CCPP) (Supplementary Table 4). Therefore, substitution of BSCF for precious Pt CE in the conventional single-SE C-P sensor can markedly enhance the sensor performance, enabling Pt CE-free sensor capable of discriminative gas detection. In comparison to state-of-the-art multivariable sensors especially the C-based ones, SnO₂-BSCF achieves discrimination of a large number of analyte gases in a relatively low concentration range with 3D dispersion (Supplementary Table 1). The benefits of MIEC SE are particularly useful for a higher-order C-P sensor with multiple regular SEs, wherein the performance of all the P-outputs can be enhanced.

## Early warning of fires

Triggered by the excellent sensing performance of SnO₂-BSCF for VOC gases, we examine the sensor practicability for fire warning. As a demonstration, we choose a common combustible, polyvinyl chloride (PVC) wire insulation, for which 2-EH is the signature gas of early hazard[35]. 0.1 g (~1 cm long) cable insulation of two representative PVC cables, one with and the other without phthalate plasticizer, were heated in the temperature range of 90-200 °C. The generated vapors were carried to the sensor chamber with a 200 ml/min air stream, which dilutes markedly the vapors, making the detection more challenging than the usual cases in a confined space. Figure 5a-b show dynamic C and P response curves of SnO₂-BSCF for the cable with phthalate (Cable 1). Significant resistance decrease for SnO₂ is observed since a cable temperature as low as 110 °C, while considerable P response starts from ~130 °C for both SEs. Comparatively, SnO₂ exhibits higher C sensitivity while BSCF performs much better in the P response to the cable vapors. At a cable temperature of 150 °C, the C

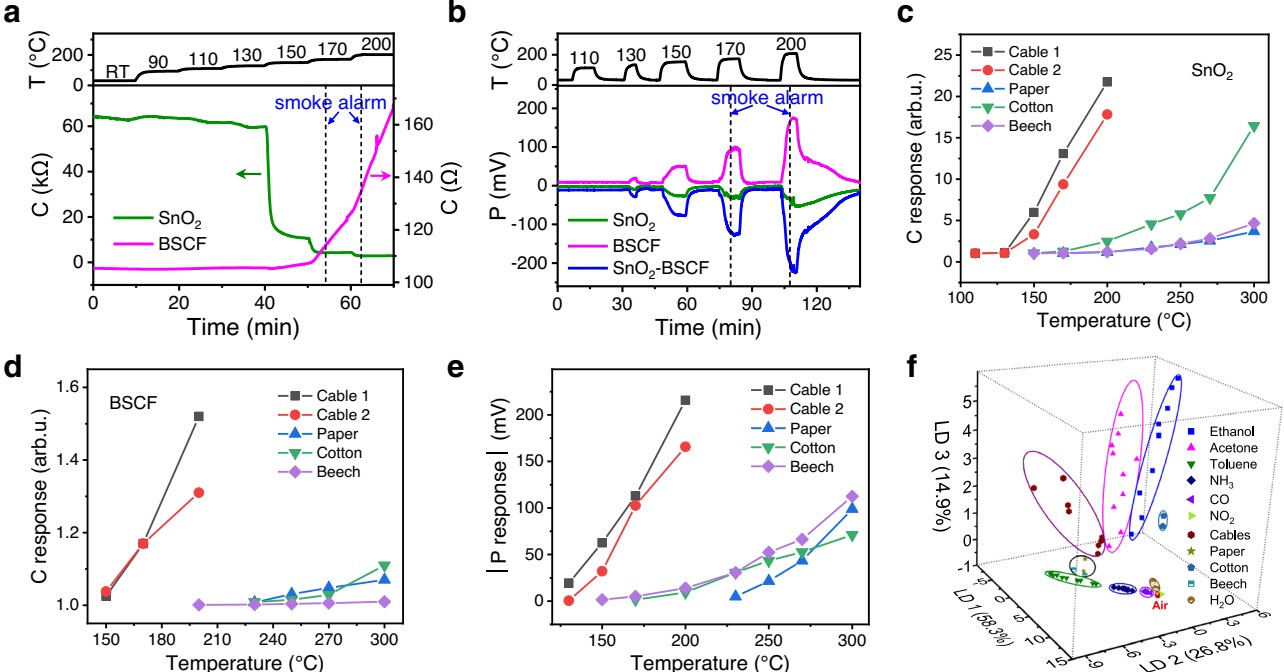

**Fig. 5 | Application of SnO₂-BSCF C-P sensor for early warning of fires.**
**a** Chemiresistive and **b** potentiometric response curves of SnO₂, BSCF, and SnO₂-BSCF sensors as a function of time at different overheating temperature of cable 1. RT in (a) represents room temperature. The green and pink solid lines in (a) represent C signals of SnO₂ and BSCF SE, respectively. The green, pink, and blue solid lines in (b) represent P signals of SnO₂ (vs Pt CE), BSCF (vs Pt CE), and SnO₂-BSCF, respectively. Left and right dotted lines in (a) and (b) represent the time of commercial smoke detector going off when the overheating temperature of Cable 1 reaches 170 °C and 200 °C, respectively. Chemiresistive response values of (**c**) SnO₂ and (**d**) BSCF and (**e**) potentiometric response values of SnO₂-BSCF sensor to 5 fire hazards (Cable 1, Cable 2, paper, cotton, and beech) as a function of overheating temperature. **f** Pattern recognition of clean air, 6 common gases in the concentration range from 20 to 100 ppm, relative humidity from 2.3% to 96.5%, and 5 fire hazards (two cable vapors obtained at 200 °C, and paper, cotton, and beech vapors obtained at 300 °C) based on LDA using the SnO₂-BSCF C-P sensor. The data variances in the projection direction of LD1, LD2, and LD3 account for 58.3%, 26.8%, and 14.9% of the total variance, respectively. Sensor operating temperature is 400 °C. Source data are provided as a Source Data file.

response is 5.97 for SnO₂ and 1.03 for BSCF, and a P response as large as −62.63 mV is obtained for SnO₂-BSCF. The sensors respond faster and more pronouncedly for higher cable temperatures because of the enhanced vapor emission. By contrast, commercial smoke detector, which is the benchmark device used nowadays for fire warning, does not go off at the cable temperature of 150 °C or below within a duration of 40 min; it alarms only after the cable has been heated at 170 °C or 200 °C for a substantial period. Similar phenomena can be observed for the phthalate-free cable (Cable 2), where the onset temperatures for the P response and the smoke alarm are slightly higher than the case of Cable 1 (Supplementary Fig. 32). It becomes manifest that SnO₂-BSCF can efficiently detect the cable overheating and is particularly suited for detection at the early stage, where it is advantageous over state-of-the-art smoke detection technique.

In practice, it is critical to reliably and accurately identify the type of fire hazard so that targeted measures can be taken without delay. We therefore compare the response of SnO₂-BSCF to vapors generated from several other common fuels, including paper, cotton, and beech. In contrast to the case of cable overheating, the responses for these substances are much smaller and start at temperatures around 70-100 °C higher (Fig. 5c−e). In other words, SnO₂-BSCF can detect cable overheating more sensitively and earlier than the other fire hazards. The discriminative capability of SnO₂-BSCF is illustrated in Fig. 5f for a more complicated case, considering 5 different fuel vapors and 7 common interfering gases. The various analytes are distinctly separated in the 3D response plot (cable vapors as a group) and distinguished from air except some overlapping of the beech and paper vapors. These findings highlight the potentials of SnO₂-BSCF C-P sensor for highly sensitive and discriminative detection of early fires and VOCs.

## Discussion

In summary, we demonstrate a gas-sensing platform with multiple C and P readouts for efficient gas discrimination. Combining the P signal is generally more advantageous to gas recognition over the C one (Supplementary Tables 2&4). A conceptual C-P sensor with dual SEs of reverse P-response polarity manifests markedly enhanced VOC detection down to ppb level, and accomplishes 3D response and gas discrimination with a higher accuracy than conventional arrays. In comparison to arrays, the multivariate C-P sensor features several advantages. Firstly, it allows to tune (reduced or enhanced depending on the need) the P response by electrode pairing, which may increase the accuracy of gas identification. Secondly, the use of precious and inefficient Pt electrode can be avoided, less materials are needed, and the device becomes more compact (Supplementary Fig. 33), diminishing the size and cost of the device. These benefits, which become more evident when a larger number of SEs are used, are highly desirable for the portability, miniaturization, and integration, and for large-scale applications in the Internet of Things era[43]. Finally, all SEs are deposited on a same substrate and operate on the same temperature. This can avoid the problem of internal heat transfer interference encountered by arrays, which usually are composed of sensors working at different temperatures[44], and thereby would help mitigate the drifting and improve stability.

Since the multivariate C-P sensor has the same functions as arrays of standalone C and P sensors, the knowledge gained and technologies developed for the C and P transducers could be smoothly applied to the C-P sensing platform. Therefore, it is possible to further improve the performance and applicability of the prototype C-P sensor. For example, the SE material may be extended to a broad range of other gas-sensitive semiconductors with diverse nanostructures, such as

carbon nanotubes, graphene, transition metal dichalcogenides, and MXenes[39]. Tuning the combinations of materials (type, composition, morphology, etc.) and operating conditions will enable to tailor and optimize the sensing pattern and discriminative capacity for customized applications, even those with stringent requirements such as concentration quantification and/or discrimination of gas mixture[12,45]. Using MEMS technology and microhotplates based on thin films of solid electrolytes, low-power miniature C-P sensors can be envisioned[46]. Combining response-regulating approaches like surface modification[7,47], catalytic or filter overlayer[48], temperature modulation[49], and light activation[50] may further enhance the sensing performance. The sensor applicability can be expanded if room-temperature operation is realized, which is feasible by using (quasi-)solid state electrolytes with high ionic conductivity at ambient temperature[51,52]. Although Pt current collectors and leads are used in this work to avoid possible sensor failure during high-temperature operation, they can be readily replaced with other metals like Au, Ag, Ti, and their alloys, which are more common in practice especially for operation at relatively low temperatures. In that case, the dual- or multi-SE C-P sensors will become Pt-free and more cost effective. On the other hand, the C-P sensor may be used as a promising solution to the humidity problem. When compared with other techniques such as material functionalization[53,54], surface coating[55], and using filters[56] or humidity sensor[14], this approach requires no additional synthesis or auxiliary parts and would be advantageous in terms of simplicity and compactness.

## Methods

### Synthesis of semiconductor sensing materials

All reagents were of analytical grade and purchased from Sinopharm Chemical Reagent Co., Ltd., China, unless otherwise stated. $SnO_2$ nanofibers (NFs) were prepared via electrospinning. In a typical process, homogenous electrospinning precursor was obtained by dissolving 0.8 g $SnCl_2 \cdot 2H_2O$ ($\geq 98\%$) and 0.8 g Polyvinyl pyrrolidone (PVP, MW = $1.3 \times 10^6$, Alfa Aesar) into 4.7 ml N,N-dimethylformamide (DMF, $\geq 99.5\%$) and 5.6 ml anhydrous ethanol ($\geq 99.7\%$) under stirring. The transparent solution was then transferred to a plastic syringe mounted on a syringe pump (LSP01-1A). Electrospinning was conducted at a voltage of 15 kV at an injection rate of 0.4 ml/h and a working distance of 15 cm. The as-spun nanofiber mats were dried at 80 °C and calcined at 600 °C for 3 h to obtain $SnO_2$ NFs. BSCF powders were synthesized by a sol-gel method. Appropriate amount of $Ba(NO_3)_2$ ($\geq 99\%$) was first dissolved in edetic acid (EDTA) ($\geq 99.5\%$)-$NH_3 \cdot H_2O$ solution under heating and stirring, then the calculated stoichiometric amounts of $Sr(NO_3)_2$ ($\geq 99.5\%$), $Co(NO_3)_2 \cdot 6H_2O$ ($\geq 98.5\%$) and $Fe(NO_3)_3 \cdot 9H_2O$ ($\geq 98.5\%$) were added the solution; citric acid ($\geq 99.5\%$) was then added to the solution under stirring to obtain a mixed solution with the mole ratio of EDTA, citric acid, and total metal ions = 1:1.5:1. $NH_3 \cdot H_2O$ was added to adjust the pH of the mixed solution to 6. The mixed solution was evaporated to dry gel under continuous stirring and heating, followed by calcination at 950 °C for 5 h to obtain BSCF powders. LSCF powders synthesized via a sol-gel route with calcination at 700 °C for a duration of 3 h were provided by Ruier Chemical Technology, China. Synthesis of NiO NFs, Fe-doped NiO NFs with nominal Fe/Ni ratio of 0.5, 3, and 10 at.% (denoted as NiO-0, NiO-0.5, NiO-3, and NiO-10, respectively), and ZnO NFs is described in the Supplementary Note 1.

### Fabrication of solid electrolytes

All reagents were of analytical grade and purchased from Sinopharm Chemical Reagent Co., Ltd., China, unless otherwise stated. GDC powders were prepared by citrate-nitrate combustion. Typically, $Ce(NO_3) \cdot 6H_2O$ ($\geq 99\%$) and $Gd_2O_3$ ($\geq 99.9\%$) were dissolved in the nitric acid solution under stirring. Citric acid was added to obtain a solution with the mole ratio of total metal ions to citrate = 1:1.5. The

pH of the solution was adjusted to 8-9 by adding $NH_3 \cdot H_2O$. The solution was heated to spontaneous combustion, and the obtained ash was calcined at 1000 °C for 3 h to prepare the GDC powders. The GDC powders of 0.6 g were subject to a uniaxial pressure of 225 MPa to obtain a disk. After sintering the disk at 1500 °C for 10 h, GDC ceramic disks of 12.5 mm diameter and 1 mm thickness were obtained. YSZ disks of 13 mm×13 mm size and 0.25 mm thickness were purchased from Hefei Kejing Co., Ltd., China. Preparation of the other solid electrolytes (ESB and NASICON) is described in Supplementary Note 2.

### Fabrication of sensors

To fabricate current collectors and CE, strip-shaped Pt pastes (PE-Pt-7840, Sino-Platinum Co., Ltd., China) were coated on the same side of the electrolyte disk, connected with Pt wires, and then sintered at 1000 °C for 30 min. To obtain SEs, slurries consisting of semiconductor sensing materials and organic binders (α-terpineol and ethyl cellulose from Shanghai Aladdin Bio-Chem Technology) were drop-cast or brush-painted on the electrolyte disk through a shadow mask and sintered under appropriate conditions (Supplementary Table 5). C-P multivariable sensors were thus obtained.

### Sensor tests and calculation

The resistance of the SE (C signal) was measured with a digital electrometer (Agilent, 34970 A), picoammeter (Keithley, 6482), or sourcemeter (Keysight, B2900). The open circuit voltage (OCV) of the multivariable sensor (P signal) was measured with the digital electrometer, whose positive terminal was connected to one of the two current collectors under the SE and negative terminal connected to CE (in the case of single SE sensors) or one of the two collectors under one other oxide SE (for dual-SE sensors). To reflect the pristine sensing performances, the C and P signals were measured individually. Detailed procedures about the sensor tests are described in the Supplementary Note 3. Depending on the type of material and analyte tested, C response is calculated from $C_g/C_a$ or $C_a/C_g$, where $C_a$ and $C_g$ are the resistance of the SE in air and analyte gas, respectively. P response is defined as the difference between the OCV in analyte gas and in air ($P_{gas}-P_{air}$). Sensitivity (S) is represented by the slope of the linear fit for the plot of dependence of response on gas concentration. LOD is calculated from the root-mean-square noise (rms) and sensitivity according to LOD = 3 × $rms/S$[8]. Response/recovery time is defined as the time for the resistance/voltage variation to achieve 90% of the total change after feeding/removal of analyte gas.

## Data availability

The source data used in this study are available in the Zenodo database under accession code https://doi.org/10.5281/zenodo.7920777[57]. Source data are provided with this paper.

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

## Acknowledgements

This work was supported by the National Key Research and Development Program of China (2021YFC3000300, W.S. and J.Y.), the National Natural Science Foundation of China (62271462, J.Y.; 61871359, J.Y.; 52204194, H.Z.), and Fundamental Research Funds for the Central Universities (WK2320000053, J.Y.).

## Author contributions

J.Y. conceived the research and supervised the project. H.Z. performed the experiments. Z.Z., Z.L., and H.H. assisted in the gas-sensing experiments and data analysis. J.Y. and H.Z. wrote the manuscript. W.S. contributed in scientific discussion. All authors reviewed and commented on the manuscript.

## Competing interests

The authors declare no competing interests.
