## [Peer Review File · Nature Communications]

Reviewer #1 (Remarks to the Author):

The manuscript of Zhang and co-workers brings an interesting and exciting potential path toward more accurate gas sensing. The idea of chemiresistive-potentiometric sensing should be of broad interest to the multidisciplinary readership of Nature Communications. The manuscript should be revised per several comments. First, several additional experiments should be performed. Second, authors should revise this manuscript to provide accurate statements. Once the manuscript is revised per these comments, it can be re-submitted for a re-evaluation.

1. Authors should add numerous literature references on two-dimensional (2D) multivariable sensors over the last 2-10 years. Authors should try to expand dimensionality of response of their multivariable sensor from 2D response to more dimensions after their LDA or PCA. They should try to achieve as least a 3D response.
2. Statement “arrays of conventional single-output sensors suffer from significant rise of size, power consumption, complexity, and cost with the number of sensors” is so misleading and should be replaced with a statement that drift problems of sensor arrays in the lab persist even after 40 years of arrays development and do not allow to be in practical uses.
3. Authors should remove word “long term” from discussion of sensor responses for 20 days and add new experimental data showing how stability of their 3D response pattern is preserved over this time of 20 days. Authors can site papers with stability studies of sensors tested for several years. Those responses are appropriate to call “long term”.
4. In abstract, after the last sentence “Our findings offer possibilities to design simple, compact, inexpensive, and highly efficient multivariate gas sensors”, authors should add their sentence from suppl. info “Test results at 96.5% RH revealed that the chemiresistive and potentiometric behaviors of SnO₂ were adversely affected by the humidity”, remove “typical for semiconductor gas sensors” because not all semiconductor sensors are affected by RH at their high operation temperature of 450-550 oC. Authors should add detailed discussion in Summary or Discussion sections on how to overcome this problem.
5. Authors should remove words “trace” and “toxic” from their text. Ethanol TWA level is 1000 ppm while NO₂ TWA is 5 ppm.
6. Authors should include dynamic raw data that was used for their calculations of the limit of detection (LOD) and a table of LOD for all tested gases.

Reviewer #2 (Remarks to the Author):

The manuscript concerns a multisensor that combines chemoresistive and potentiometric sensors to detect and distinguish different gases, VOCs and odors emitted by heated materials. The structure of the manuscript is consistent and the results are shown in great detail, with a very long and detailed Supplementary Material. Overall the manuscript is good, although there are a couple of problems such as the lack of sensor images and some explanations that are not easy for the non-specialist reader.

Some more specific comments are given in the following lists.

Major points:

- How big are the sensors? How does the deposition of the different materials take place in a localized way? It would be important to insert some SEM images of the various sensors.
- There are already multiple sensors or miniaturized electronic noses that have performance comparable or better to that of the multisensor presented in this manuscript (often they are also able to estimate the gas concentration).

For example: 10.1021 / acssensors.1c01204 - 10.1016 / j.aca.2020.05.015 - 10.1038 / s41598-017-10495-8 - 10.1016 / j.snb.2012.02.095

How do the authors explain that their device is better than those already made?

- Lines 334-337: "To fabricate multivariable sensing platform, strip-shaped Pt pastes (Sino-Platinum Co., Ltd., China, 1 mm×3 mm) were coated on the same side of the electrolyte disk and Pt wires were connected to the Pt strips with small point shaped Pt pastes..." is in contrast to what is written in Lines 127-132, in which the authors indicate their sensor as platinum-free (which they also do elsewhere in the manuscript).

Minor points:

- It is best, for the reader's understanding, to maintain uniformity within the manuscript. For example, in Fig. S6 the sensors are called SnO₂-R and SnO₂-V (I guess as Resistance and Voltage), while in the text (for example in Line 134) they are usually called C (SnO₂) and P (SnO₂-LSCF), with C from Chemiresistive and P from Potentiometric.
- Perhaps it would be useful to use the scale of the P answer by reversing it. For example, in Fig. S7, if the scale were reversed it could be more easily understood that the trend is similar for all four plots in Fig. S7a and b. This also applies to other images throughout the manuscript, but mine is just a suggestion.
- Line 105: "Er_{0.4}Bi_{1.6}O_{3-δ}" should probably be "Er_{0.4}Bi_{1.6}O_{3-δ}"
- Line 143 and Supplementary Note 13: The four sensors used are not defined. It is not clear what the names NiO-0, NiO-0.5, NiO-3, and NiO-10 refer to. Probably to a different doping/surface decoration with Fe, but I have not found the explanation anywhere.
- Fig. S20: the figure caption seems to report the wrong sensor names.
- There are some syntax errors, but they are minor and do not affect understanding.
- Lines 192-193: "Such unusual behavior is a unique feature that can be utilized to enhance the sensor performance." Why? A negative correlation is as good as a positive correlation between sensor responses to achieve selectivity.
- Lines 197-198: It seems strange that the LOD is 26 ppb by observing the noise of the signal in the inset of Fig. 3c.
- Line 200: Again homogeneity: here it is called R response, while a few lines later it is called C response.
- Fig. 3f and others: How do the authors explain that strange linearity between the points of some different gases?
- Fig. 5f: normal air is not present in the graph, so how do we know how distinct the gases that indicate fire are compared to air?

Reviewer #3 (Remarks to the Author):

It is noteworthy that authors developed a novel approach, in which chemiresistive and potentiometric gas sensing have been combined.

Prior to evaluation, I have some inquiries:

1] Please add references on combined approaches [chemiresistive and potentiometric] in more detail.

2] Please state the interfering effects of both sensors in more detail. Does the combination degrade the sensing performances of each component?

● Reviewer #1

The manuscript of Zhang and co-workers brings an interesting and exciting potential path toward more accurate gas sensing. The idea of chemiresistive-potentiometric sensing should be of broad interest to the multidisciplinary readership of Nature Communications. The manuscript should be revised per several comments. First, several additional experiments should be performed. Second, authors should revise this manuscript to provide accurate statements. Once the manuscript is revised per these comments, it can be re-submitted for a re-evaluation.

General response: We thank the reviewer for the positive feedbacks and the valuable comments that of great help in improving the manuscript. Our point-to-point response is detailed below.

1. Authors should add numerous literature references on two-dimensional (2D) multivariable sensors over the last 2-10 years. Authors should try to expand dimensionality of response of their multivariable sensor from 2D response to more dimensions after their LDA or PCA. They should try to achieve as least a 3D response.

Response: We highly appreciate these constructive and useful comments. More literature references on multivariable sensors with demonstrated gas discrimination have been added into the manuscript (Ref 24-27) and the Supplementary Table 1 (Ref 3, 5, 7, 10, 13-14, 16-17, 19, 22-24)

With respect to the dimensionality of response, we did achieve 3D response with our dual-SE C-P sensors with the obtained three signals (CCP), since the number of

independent outputs corresponds with the dimensionality. Unfortunately, the corresponding LDA patterns were plotted in a 2D mode in the original manuscript. We have plotted the LDA patterns for the dual-SE C-P sensors in a 3D mode in the revised Fig. 2b, 4c, and 5f. These 3D plots indeed show more clearly the separation of the gases than the previous 2D ones. Higher-order response is in principle achievable by using more SEs. As an example, we obtain 7 outputs with 4 SEs based on Fe-doped NiO nanofibers (Supplementary Fig. 18), i.e., 4 C and 3 P signals, which are the basis for 7D response.

Rev. Fig. 2b. LDA pattern recognition of 3D response to 6 different gases using the SnO₂-LSCF C-P sensor.

Rev. Fig. 4c. LDA pattern recognition of 3D response to 7 different gases and humidity using the SnO₂-BSCF C-P sensor.

Rev. Fig. 5f. LDA pattern recognition of 3D response to 4 different fuel vapors, 6 common interfering gases, humidity, and clean air using the SnO₂-BSCF C-P sensor.

2. Statement “arrays of conventional single-output sensors suffer from significant rise of size, power consumption, complexity, and cost with the number of sensors” is so misleading and should be replaced with a statement that drift problems of sensor arrays in the lab persist even after 40 years of arrays development and do not allow to be in practical uses.

Response: Many thanks for pointing out the inaccurate expression. Indeed, drift has been one major problem limiting practical uses of sensor arrays. On the other hand, using a larger number of standalone sensors in arrays for higher-order sensing and better accuracy is inevitably associated with a rise of size and cost, which is unfavorable for practical applications especially when portability and miniaturizability are demanded.

We have made these clear in the manuscript (lines 31-34, page 2).

3. Authors should remove word “long term” from discussion of sensor responses for 20 days and add new experimental data showing how stability of their 3D response pattern is preserved over this time of 20 days. Authors can cite papers with stability studies of sensors tested for several years. Those responses are appropriate to call “long term”.

Response: We appreciate the invaluable suggestions. The reproduction of the 3D response pattern relies mainly on the performance repeatability of the C-P sensor, which retains the characteristics of both conventional C and P sensors (see Fig. 1). The stability of C and P sensors is jointly affected by various factors, such as the nature of component materials, microstructure, fabrication techniques and processes, and working conditions. Optimizing these factors can greatly improve the sensor stability, and may achieve stable operation for years (J Electrochem Soc 164, 2017, B681 and Sens Actuators B Chem 143, 2009, 246). Commercial chemiresistive / potentiometric sensors (e.g. Figaro methane sensor / Bosch oxygen sensor) have performed quite well in this respect over the past decades. Therefore, similar long-term stability should likely be achieved for the C-P sensor under appropriate conditions.

The main goal of present work is the demonstration of the C-P sensing platform for enhanced discriminative gas detection. As the complete response/recovery especially for the VOC gases is rather sluggish, a complete test cycle to obtain the 3D response pattern lasts over ~25 days. In order to achieve our goal efficiently, we exemplify the reliability of our sensor test and stability of materials especially BSCF as a novel electrode with the response of P(SnO₂), P(BSCF), and P(SnO₂-BSCF) to 2-EH. We

intermittently measured these responses over a period of 52 days, during which the data for the 3D response pattern in Fig. 4c were collected. Results reveal excellent reversibility and reproducibility of the response over the test cycle (Supplementary Figs. 30-31), which also reflect stability of the materials. We reason that the response to other gases and resultant 3D pattern can similarly be reproduced in this work.

We have removed the word “long term” and revised relevant expressions in the manuscript (lines 227-231, page 10) and Supplementary Fig. 31.

Rev. Supplementary Fig. 31. Stability test of the potentiometric response to 100 ppm 2-EH for the SnO₂-BSCF sensor at 400 °C. Data for SnO₂ and BSCF sensors are also shown for comparison.

4. In abstract, after the last sentence “Our findings offer possibilities to design simple, compact, inexpensive, and highly efficient multivariate gas sensors”, authors should add their sentence from suppl. info “Test results at 96.5% RH revealed that the chemiresistive and potentiometric behaviors of SnO₂ were adversely affected by the humidity”, remove “typical for semiconductor gas sensors” because not all

semiconductor sensors are affected by RH at their high operation temperature of 450-550 °C. Authors should add detailed discussion in Summary or Discussion sections on how to overcome this problem.

Response: Many thanks for addressing this issue, which help improve the clarity of the work. We agree that RH is often a nuisance for semiconductor gas sensors, which is not always the case particularly at relatively high temperatures. The emphasis of this work is placed on the illustration of the features and advantages (such as enhanced discrimination) of the C-P sensor. As shown in revised Fig. 4c and 5f, a single SnO₂-BSCF C-P sensor can discriminate RH from other gases, despite significant/moderate effect of RH on both C and P signals of SnO₂/BSCF. Hence, the C-P sensor could possibly be used as a promising solution to the humidity problem. When compared with existing techniques such as material functionalization, surface coating, and using filters or humidity sensor, this approach requires no additional synthesis or auxiliary parts and would be advantageous in terms of the simplicity and compactness.

We have made the above points clear in the Discussion section of the manuscript (lines 316-320, page 14) and removed the statement “typical for semiconductor gas sensors”.

5. Authors should remove words “trace” and “toxic” from their text. Ethanol TWA level is 1000 ppm while NO₂ TWA is 5 ppm.

Response: Thanks for the comments. We have deleted the expressions "trace" and "toxic" in the revised manuscript.

6. Authors should include dynamic raw data that was used for their calculations of the limit of detection (LOD) and a table of LOD for all tested gases.

Response: We appreciate the constructive suggestion. The dynamic raw data used for LOD calculations (Supplementary Fig. 29) and a table of LOD (Supplementary Table 3) for all tested gases have been included in Supplementary materials. Relevant discussions have been added in the manuscript (lines 221-226, pages 9-10).

Rev. Supplementary Table 3. LOD data of P(SnO₂-BSCF) sensor for 7 target gases.

Gas	LOD (ppm)	TWA (ppm) *
2-EH	0.00811	1
C ₇ H ₈	0.05591	50
CH ₃ CH ₂ OH	1.39	1000
CH ₃ COCH ₃	0.85	500
NH ₃	1.65	20
CO	1.53	20
NO ₂	1.28	0.5

*: Occupational Exposure Limits of European Union or OSHA (ethanol).

● Reviewer #2

The manuscript concerns a multisensor that combines chemoresistive and potentiometric sensors to detect and distinguish different gases, VOCs and odors emitted by heated materials. The structure of the manuscript is consistent and the results are shown in great detail, with a very long and detailed Supplementary Material. Overall the manuscript is good, although there are a couple of problems such as the lack of sensor images and some explanations that are not easy for the non-specialist reader.

Some more specific comments are given in the following lists.

General response: We thank the reviewer for the positive assessment of our work and constructive comments. Our point-to-point response is detailed below.

Major points:

1. How big are the sensors? How does the deposition of the different materials take place in a localized way? It would be important to insert some SEM images of the various sensors.

Response: Thanks for raising these points that help improve the clarity of our manuscript. For the sake of easy comparison and convenience, our sensors were made on GDC disks with a diameter of 12.5 mm, which can accommodate up to 4 SEs with a diameter of 3 mm. The different SEs were deposited onto the GDC by drop-casting or brush-painting the corresponding slurries through a shadow mask. Details about the sensor deposition along with (SEM) images and schematic drawings of representative sensors have been added to the revised manuscript (lines 337-340, page 15) and

Supplementary Materials (Supplementary Figs. 2, 21, 25).

Rev. Supplementary Fig. 2. (a) Photograph and (b) schematic of the SnO₂ C-P sensor; surface SEM image of (c) SnO₂ nanofibers SE and (d) GDC electrolyte disk.

Rev. Supplementary Fig. 21. (a) Surface and (b) cross-section SEM image of BSCF SE.

Rev. Supplementary Fig. 25. (a) Photograph and (b) schematic of the dual-SE SnO₂-BSCF C-P sensor.

2. There are already multiple sensors or miniaturized electronic noses that have performance comparable or better to that of the multisensor presented in this manuscript (often they are also able to estimate the gas concentration). For example: 10.1021 / accsensors.1c01204 - 10.1016 / j.aca.2020.05.015 - 10.1038 / s41598-017-10495-8 - 10.1016 / j.snb.2012.02.095. How do the authors explain that their device is better than those already made?

Response: We really appreciate these insightful and constructive comments, which help improve clarity of the manuscript.

Conventional array (the stated multiple sensor) is typically a simple combination of various single-output gas sensors of the same type (e.g., chemiresistive ones in the aforementioned references 10.1021 / accsensors.1c01204, 10.1016 / j.aca.2020.05.015, and 10.1038 / s41598-017-10495-8) or different types (chemiresistive and capacitive ones in 10.1016 / j.snb.2012.02.095). As each component sensor is a standalone and has different materials and operating temperatures, the array is intrinsically associated

with problems such as uncorrelated drift, large size, and high cost (see Response to Reviewer 1's comment No. 2). These problems could be alleviated or overcome with a multivariate sensor, which outputs independent signals of different working principles from the same sensing material.

Our C-P sensor retains the sensing characteristics of both conventional C and P sensors (Fig. 1). A direct comparison of our multivariate sensor with reported arrays is not straightforward, given the different conditions (sensor amounts, gases, concentration ranges, etc.). Instead, it would be appropriate and fair to compare the multivariate C-P sensor and the corresponding counterpart arrays of standalone C and P sensors (such as CC, CCP, CPP, CCPP), using the same materials and working conditions. Such comparison reveals that the C-P sensor can not only fulfill the functions of arrays but also offer additional advantages in several aspects. Firstly, the P response of a C-P sensor can be tuned (reduced or enhanced depending on the need) by using different SE combinations, which can generate a more diverse sensing pattern, thereby increasing the accuracy of gas identification (see Fig. 2 & 4 and Supplementary Tables 2 & 4). Secondly, the use of precious and inefficient Pt electrode can be avoided, less materials are needed, and the device becomes more compact, diminishing the size and cost of the device. These benefits are highly desirable for the portability, miniaturization, and integration, and for large-scale applications in the Internet of Things era, which become more evident when a larger number of sensing SEs are used (Supplementary Fig. 33). Finally, all SEs of the C-P sensor are deposited on a same substrate and operate at the same temperature. This is helpful to resolve the problem of

internal heat transfer interference encountered by conventional arrays, whose component sensors may work at different temperatures, and thereby would help mitigate the drifting and improve stability. As the responses scale with gas concentration and the signals differ in the sensitivity for a same analyte, the C-P sensor will also be able to determine the gas concentrations (Fig. R1), like other multivariate gas sensors and arrays.

We have supplemented the above-mentioned references in the Introduction section and summarized the advantages of the C-P sensor in the Discussion section (lines 292-306, pages 13-14), which also strengthens the advance of the work.

Rev. Supplementary Fig. 33. Comparison of conventional arrays and corresponding C-P sensors that output (a) 3 and (b) 7 different independent C/P signals. The C-P sensors, outputting the same signals as arrays whilst performing better, use less materials (including substrates, Pt, and sensing materials) and are smaller and more compact.

Fig. R1. Exemplary LDA plot for three gases of varied concentrations for the single-SE C-P BSCF sensor.

3. Lines 334-337: “To fabricate multivariable sensing platform, strip-shaped Pt pastes (Sino-Platinum Co., Ltd., China, 1 mm×3 mm) were coated on the same side of the electrolyte disk and Pt wires were connected to the Pt strips with small point shaped Pt pastes...” is in contrast to what is written in Lines 127-132, in which the authors indicate their sensor as platinum-free (which they also do elsewhere in the manuscript).

Response: Many thanks for pointing out these inconsistent and inaccurate statements, and we are sorry for the confusion they aroused. Our sensors use Pt pastes and wires for current collection and connection, and thus are not platinum free. In practice other less expensive metals like Au, Ag, Ti, and their alloys function well and are more often used for current collection and connection at relatively lower temperatures, below ~400°C. Nevertheless, they may have stability issue at relatively high temperatures, which would cause interference or failure to the sensor test. In this work, some tests were conducted at temperatures above 400°C intended to demonstrate the successful

operation of the C-P sensor over a wide temperature range. We thus chose the high temperature-resistant Pt as current collectors and connection to avoid possible stability problem.

Conventional P sensors commonly use Pt as counter electrode (CE). In contrast, the Pt CE can be replaced by an oxide SE in a C-P sensor with two or more SEs. Thus, our dual- and four-SE C-P sensors are free of Pt CE, which reduces the Pt usage and sensor cost. When other metals are used for current collection and connection as discussed above, the dual- or multi-SE C-P sensors will become real Pt-free ones.

We have revised the “Pt-free sensors” to “Pt CE-free sensor” in the manuscript to make this point clear.

Minor points:

4. It is best, for the reader's understanding, to maintain uniformity within the manuscript. For example, in Fig. S6 the sensors are called SnO₂-R and SnO₂-V (I guess as Resistance and Voltage), while in the text (for example in Line 134) they are usually called C (SnO₂) and P (SnO₂-LSCF), with C from Chemiresistive and P from Potentiometric.

Response: We thank the reviewer for the useful suggestions. We have uniformed the related expressions throughout the manuscript and Supplementary Materials, with C for Chemiresistive and P for Potentiometric.

5. Perhaps it would be useful to use the scale of the P answer by reversing it. For example, in Fig. S7, if the scale were reversed it could be more easily understood that the trend is similar for all four plots in Fig. S7a and b. This also applies to other images throughout the manuscript, but mine is just a suggestion.

Response: We appreciate the reviewer's constructive suggestions. For better clarity and consistency, we have followed the suggestions and used the absolute value of the P response in relevant figures throughout the revised manuscript and Supplementary Material for all samples except for BSCF whose P response is in itself positive.

6. Line 105: "Er_{0.4}Bi_{1.6}O_{3-δ}" should probably be "Er_{0.4}Bi_{1.6}O₃-δ".

Response: We apologize for the typo. We have corrected Er_{0.4}Bi_{1.6}O_{3-δ} to Er_{0.4}Bi_{1.6}O₃-δ in the revised version.

7. Line 143 and Supplementary Note 13: The four sensors used are not defined. It is not clear what the names NiO-0, NiO-0.5, NiO-3, and NiO-10 refer to. Probably to a different doping/surface decoration with Fe, but I have not found the explanation anywhere.

Response: Thanks for pointing these out. NiO-0, NiO-0.5, NiO-3 and NiO-10 stand for Fe-doped NiO NFs with nominal Fe/Ni ratio of 0, 0.5, 3, and 10 at.%, respectively. We have explained the definitions in the manuscript (lines 325-326, page 14) and provided the actual Fe contents in Supplementary materials.

8. Fig. S20: the figure caption seems to report the wrong sensor names.

Response: We apologize for the mistake. Wrong sensor names were given in the legend of former Supplementary Fig. 20 (present Supplementary Fig. 19), which have been corrected.

9. There are some syntax errors, but they are minor and do not affect understanding.

Response: We thank the reviewer for this comment. We have carefully checked and improved the grammar and writing throughout the manuscript.

10. Lines 192-193: “Such unusual behavior is a unique feature that can be utilized to enhance the sensor performance.” Why? A negative correlation is as good as a positive correlation between sensor responses to achieve selectivity.

Response: Thanks a lot for pointing out this ambiguity. Indeed, BSCF will function similarly to conventional materials if it is paired with Pt electrode in a single-SE sensor. The advantages of the unusual polarity are manifested when BSCF is paired with a conventional SE to form a dual-SE sensor, leading to enhanced performance due to their opposite polarity, as shown in Fig. 4b-c, Supplementary Fig. 28, and Supplementary Table 4 and elaborated in the section “Enhancing the multivariable sensing performance” in the manuscript.

We have removed the above-mentioned sentence from Section “Mixed conducting perovskite oxides as high-performance SE” in the revised manuscript.

11. Lines 197-198: It seems strange that the LOD is 26 ppb by observing the noise of the signal in the inset of Fig. 3c.

Response: Thanks for raising this issue. According to the method of IUPAC: $LOD = 3rms/S$, where rms is the root mean square noise and S is the sensitivity. The rms is 0.022 mV, and the calculated LOD is as low as 0.0266 ppm (26.6 ppb) for 2-EH, benefiting from the large S (2.5 mV/ppm). The voltage change for the LOD ($=3rms$) is only 0.066 mV, and appears to be close to the noise in the inset of Fig. 3c because of the much larger scale for the y-axis (1.5 mV). We have emphasized the importance of sensitivity to LOD in the revised manuscript (lines 223-224, page 9).

12. Line 200: Again homogeneity: here it is called R response, while a few lines later it is called C response.

Response : Thanks for pointing out this inconsistency. We have uniformed the expressions with C for chemiresistive and P for potentiometric in the revised manuscript and Supplementary Materials.

13. Fig. 3f and others: How do the authors explain that strange linearity between the points of some different gases?

Response: We thank the reviewer for the careful observation. The linearity of data points for NH_3 , C_7H_8 , H_2O , CO and NO_2 in the LDA plots like Fig. 3f is a result of the low or negligible chemiresistive sensitivity of BSCF to these gases (C response is ~ 1) in the LDA calculation. The reason is as follows.

Supposing that the 2D response output is $\begin{pmatrix} C \\ P \end{pmatrix}$, the coordinate transformation matrix is $\begin{pmatrix} a1 & a2 \\ b1 & b2 \end{pmatrix}$, and the new coordinates after transformation in the LDA plot are $\begin{pmatrix} LD1 \\ LD2 \end{pmatrix}$, then $\begin{pmatrix} a1 & a2 \\ b1 & b2 \end{pmatrix} \begin{pmatrix} C \\ P \end{pmatrix} = \begin{pmatrix} LD1 \\ LD2 \end{pmatrix}$, where $LD1 = a1 * C + a2 * P$ and $LD2 = b1 * C + b2 * P$. When C is constant (e.g., =1), the coordinate differences between any two data points are $\Delta LD1 = a2 * \Delta P$ and $\Delta LD2 = b2 * \Delta P$. Thus, $\Delta LD1 / \Delta LD2 = a2 / b2$ is a constant, resulting in linearity of data points. Conversely, neither the C nor the P response $\begin{pmatrix} C \\ P \end{pmatrix}$ of BSCF to 2-EH, ethanol, and acetone is constant, and therefore no linearity can be observed in the LDA plot for them.

14. Fig. 5f: normal air is not present in the graph, so how do we know how distinct the gases that indicate fire are compared to air?

Response: Thanks for raising this concern, which helps improve the clarity of the manuscript. We have included air as a reference in Fig. 5f, which is distinctly separated from fire signature gases. We have also made this point clear in the text (lines 277-280, page 12).

Rev. Fig. 5f. LDA pattern recognition of 3D response to 4 different fuel vapors, 6 common interfering gases, humidity, and clean air using the SnO₂-BSCF C-P sensor.

● Reviewer #3

It is noteworthy that authors developed a novel approach, in which chemiresistive and potentiometric gas sensing have been combined.

General response: We thank the reviewer for the positive assessment of the novelty of our work and valuable comments. Our point-to-point response is detailed below.

1. Please add references on combined approaches [chemiresistive and potentiometric] in more detail.

Response: We thank the reviewer for the constructive suggestion. According to our recent literature survey, chemiresistive transducer has been combined with other kinds like volumetric, thermoelectric, optical, capacitive, work function, and mass, while the potentiometric one has not yet been combined with any others. This highlights the novelty and advance of our approach. We have added more references on chemiresistive-based multivariate sensors with demonstrated gas discrimination into the manuscript (Ref. 24-27) and the Supplementary Table 1 (Ref. 3, 5, 7, and 10), and revised relevant descriptions in the manuscript (lines 46-52, page 2).

2. Please state the interfering effects of both sensors in more detail. Does the combination degrade the sensing performances of each component?

Response: We appreciate the insightful comments. Potential cross-talk or performance changes due to combination of the C and P signals as the C-P sensor is discussed below.

Our C-P sensor has the same structure and configuration as conventional P sensor,

except the presence of one extra Pt current collectors underneath the SE (see Fig. 1a and 4a). Thus, the P performance should in principle not be significantly affected by the signal combination. Regarding the C performance, the main difference between our sensor and conventional chemiresistor is that the former is based on a solid electrolyte substrate, which is conductive to solid ions like O^{2-} or Na^+ but basically non-conductive to electrons, whereas the latter is made on insulating substrate like Al_2O_3 and SiO_2 . The solid electrolyte forms a parallel connection with the sensing electrode and could thus affect the C measurement when its resistance is small. To assess potential effect of the solid electrolytes, we compare the C response of SnO_2 nanofibers on YSZ, GDC, and ESB substrates with that on Al_2O_3 . Fig. R2 shows that at relatively low temperatures ($300^\circ C$ and $400^\circ C$), where the solid electrolytes has a relatively low conductivity and large resistance, the C responses are quite close to each other. Nevertheless, at a relatively high temperature of $500^\circ C$, the C responses of the solid electrolyte-based samples are greatly reduced, which can be ascribed to significant increase of the electrolyte ionic conductivity at this high temperature. Therefore, for temperatures of $400^\circ C$ or below where most measurements in this work are conducted and higher C and P responses are obtained, combination of the C and P signals does not significantly degrade the C and P responses.

Fig. R2. Chemiresistive response of SnO₂ nanofibers (a) on different substrates to 400 ppm ethanol, (b) on Al₂O₃ and GDC substrates to 100 ppm ethanol, acetone, and hydrogen at 400°C.

Reviewer comments, second round

Reviewer #1 (Remarks to the Author):

Excellent revisions !

I suggest that the manuscript should be accepted

Reviewer #2 (Remarks to the Author):

In my opinion the manuscript is clear and well structured, but it is too specialized and not innovative enough for Nature Communications.

The work is only partially innovative and only conceptually, since:

1) There are already electronic noses that use sensors of different types:

a. D.V. Del Orbe Henriquez et al., Low-Power, Multi-Transduction Nanosensor Array for Accurate Sensing of Flammable and Toxic Gases, *Small Methods* 7 (2023) 2201352. Doi: 10.1002/smt.202201352

b. A. Oprea et al., Environmental monitoring with a multisensor platform on polyimide foil, *Sensors and Actuators B: Chemical* 171–172 (2012) 190–197. Doi: 10.1016/j.snb.2012.02.095
(these are just two examples I've found, but there are more)

2) There are electronic noses with much smaller dimensions and better performance:

a. N.X. Thai et al., Multi gas sensors using one nanomaterial, temperature gradient, and machine learning algorithms for discrimination of gases and their concentration, *Analytica Chimica Acta* 1124 (2020) 85–93. Doi: 10.1016/j.aca.2020.05.015

The device illustrated in this manuscript has potential, but it is still in the prototype stage, still "large", still containing platinum, still capable only of classifying gases but not quantifying them. In view of the partial innovation and limited performance, it is my opinion that the manuscript is not suitable for Nature Communications but for a specialist journal on sensors.

● **Reviewer #1**

Excellent revisions !

I suggest that the manuscript should be accepted

Response: We are grateful for the reviewer's positive recommendation in acceptance of our manuscript and insightful comments that help greatly improve the quality of the manuscript.

● Reviewer #2

In my opinion the manuscript is clear and well structured, but it is too specialized and not innovative enough for Nature Communications.

The work is only partially innovative and only conceptually, since:

1) There are already electronic noses that use sensors of different types:

a. D.V. Del Orbe Henriquez et al., Low-Power, Multi-Transduction Nanosensor Array for Accurate Sensing of Flammable and Toxic Gases, *Small Methods* 7 (2023) 2201352.

Doi: 10.1002/smt.202201352

b. A. Oprea et al., Environmental monitoring with a multisensor platform on polyimide foil, *Sensors and Actuators B: Chemical* 171 - 172 (2012) 190-197. Doi: 10.1016/j.snb.2012.02.095

(these are just two examples I've found, but there are more)

2) There are electronic noses with much smaller dimensions and better performance:

a. N.X. Thai et al., Multi gas sensors using one nanomaterial, temperature gradient, and machine learning algorithms for discrimination of gases and their concentration, *Analytica Chimica Acta* 1124 (2020) 85-93. Doi: 10.1016/j.aca.2020.05.015

The device illustrated in this manuscript has potential, but it is still in the prototype stage, still "large", still containing platinum, still capable only of classifying gases but not quantifying them.

In view of the partial innovation and limited performance, it is my opinion that the manuscript is not suitable for Nature Communications but for a specialist journal on sensors.

Response: We are grateful for reviewer's assessment of our manuscript and valuable comments. Below we address reviewer's concern about the novelty and advance of our multivariate C-P sensor in comparison with arrays (electronic noses) in four aspects. First of all, the advantages and importance of multivariate gas sensors in comparison with conventional arrays have been well documented and demonstrated in previous studies (see Ref. 18-20 and those listed in Supplementary Table 1). In short, multivariate gas sensors function similarly to arrays, but are advantageous in compactness, cost-effectiveness, and stability. Secondly, direct comparison of our multivariate C-P sensor with arrays of standalone C and P sensors based on the same materials and working conditions indeed reveals that the C-P sensor has better performance, reduced size, and potentials of lower cost (Supplementary Fig. 33 and Supplementary Table 4). Nevertheless, performance comparison between the C-P sensor and arrays based on other sensor types (such as those reported) is tricky, since the sensing performance of individual sensors and thus of arrays is jointly determined by and could vary markedly with many factors, especially the operating conditions and the materials. To obtain meaningful results and draw unambiguous conclusions, transducers with different working principles should be compared under the same conditions, such as the same material composition and microstructure (morphology and thickness, etc.) and temperatures. This will be practically difficult or even impossible to manage, e.g., when comparing the C and P sensors with the calorimetric/capacitive ones described in the above Ref. 1a-b because of the difference in material type (semiconductors vs metals/dielectrics) and working temperature. Careful and systematic investigation is

required and may be considered in the future. Thirdly, this work is mainly focused on the demonstration of the concept and advantages of the multivariate C-P sensing platform. The C-P sensors have the same functions as arrays of standalone C and P sensors. Hence, the knowledge gained and technologies developed for standalone C and P sensors as well as arrays based on them (such as the temperature gradient method described in the above Ref. 2) could be smoothly applied to the C-P sensing platform. This offers numerous possibilities to further improve the sensor performance and applicability, despite the prototype stage of the C-P sensors presented in this work. Since miniature Pt-free C and P sensors/arrays with capability of quantification and/or discrimination of gas mixture can be realized, this can also be achieved in a properly designed and optimized multivariate C-P sensor, which is expected to have better performance, smaller size, and lower cost. Finally, the multivariate C-P sensor desirably combines two most widely-used transducers that both are simple, low cost, and miniaturizable and have good compatibility. In contrast to most other multivariate sensors especially the C-based ones (Supplementary Table 1), which have only 2D response and/or limited applicability and high cost/complexity, the C-P sensor achieves discrimination of a large number of analyte gases in a relatively low concentration range with 3D dispersion. Higher dimensionality of response can be obtained by equipping more SEs. The higher-order response, broad applicability, low cost, and simplicity of the C-P sensing platform are highly desirable for IoT applications of large-scale sensor deployment and that demand sensors with integratability and high performance. The above points have been made clear in Introduction and Discussion. More discussions (page 10) and references including the above Ref. 1-2 are added.

Overall, our work demonstrates a multivariate gas sensing platform that opens up numerous possibilities to significantly advance the gas sensor technology and applications. We believe that it contains sufficient novelty and significance and is of broad interest to the multidisciplinary readership of Nature Communications.

Thank you again for your inspiring and constructive comments/suggestions, which help greatly improve the quality of the manuscript.